# TbpB^Y167A^-Based Vaccine Can Protect Pigs against Glässer’s Disease Triggered by *Glaesserella parasuis* SV7 Expressing TbpB Cluster I

**DOI:** 10.3390/pathogens11070766

**Published:** 2022-07-04

**Authors:** Simone Ramos Prigol, Rafaela Klein, Somshukla Chaudhuri, Gabriela Paraboni Frandoloso, João Antônio Guizzo, César Bernardo Gutiérrez Martín, Anthony Bernard Schryvers, Luiz Carlos Kreutz, Rafael Frandoloso

**Affiliations:** 1Laboratory of Microbiology and Advanced Immunology, Faculty of Agronomy and Veterinary Medicine, University of Passo Fundo, Passo Fundo 99052-900, Rio Grande do Sul, Brazil; simoneramosprigol@gmail.com (S.R.P.); rafaelaluiza.vet@gmail.com (R.K.); lckreutz@upf.br (L.C.K.); 2AFK Imunotech Ltd.a, Passo Fundo 99050-144, Rio Grande do Sul, Brazil; gparaboni@hotmail.com (G.P.F.); joaoaguizzo@gmail.com (J.A.G.); 3Department of Microbiology and Infectious Diseases, Faculty of Medicine, University of Calgary, Calgary, AB T2N 4N1, Canada; schaudhu@ucalgary.ca (S.C.); schryver@ucalgary.ca (A.B.S.); 4Departamento de Sanidad Animal, Facultad de Veterinaria, Universidad de León, Campus de Vegazana s/n, 24007 Leon, Spain; cbgutm@unileon.es

**Keywords:** *Glaesserella parasuis*, Glässer’s disease, serovar 7, vaccine, TbpB^Y167A^, cross-protection

## Abstract

*Glaesserella parasuis* is the etiological agent of Glässer’s disease (GD), one of the most important diseases afflicting pigs in the nursery phase. We analyzed the genetic and immunological properties of the TbpB protein naturally expressed by 27 different clinical isolates of *G. parasuis* that were typed as serovar 7 and isolated from pigs suffering from GD. All the strains were classified as virulent by LS-PCR. The phylogenetic analyses demonstrated high similarity within the amino acid sequence of TbpB from 24 clinical strains all belonging to cluster III of TbpB, as does the protective antigen TbpB^Y167A^. Three *G. parasuis* isolates expressed cluster I TbpBs, indicating antigenic diversity within the SV7 group of *G. parasuis*. The antigenic analysis demonstrated the presence of common epitopes on all variants of the TbpB protein, which could be recognized by an in vitro analysis using pig IgG induced by a TbpB^Y167A^-based vaccine. The proof of concept of the complete cross-protection between clusters I and III was performed in SPF pigs immunized with the TbpB^Y167A^-based vaccine (cluster III) and challenged with *G. parasuis* SV7, strains LM 360.18 (cluster I). Additionally, pigs immunized with a whole-cell inactivated vaccine based on *G. parasuis* SV5 (Nagasaki strain) did not survive the challenge performed with SV7 (strain 360.18), demonstrating the absence of cross-protection between these two serovars. Based on these results, we propose that a properly formulated TbpB^Y167A^-based vaccine may elicit a protective antibody response against all strains of *G. parasuis* SV7, despite TbpB antigenic diversity, and this might be extrapolated to other serovars. This result highlights the promising use of the TbpB^Y167A^ antigen in a future commercial vaccine for GD prevention.

## 1. Introduction

*Glaesserella parasuis* is a Gram-negative bacterium commonly found on the upper respiratory tract of pigs [1]. *G. parasuis* is transmitted mostly from sows to piglets [2] and then further spread to commingling piglets during the nursery phase. Colostrum-derived maternal antibodies provide neonatal protection during the first weeks of life [3]; however, as maternal-derived antibodies decline [4], the bacteria might overcome innate immunity, reaching target tissues, and causing a variety of clinical signs such as polyarthritis, polyserositis, meningitis and pneumonia [5], which are usually fatal.

Outbreaks of GD are usually associated with highly virulent strains, which can act as the primary disease causative agent, as suggested by our group [5]. Moderately virulent strains can take advantage of immune-suppression events caused by stress or bacterial and viral coinfections and cause disease [6,7]. Non-virulent strains are not associated with GD and their role as part of the respiratory microbiota is not well understood. Although *G. parasuis* can be classified into 15 well-known serovars (SVs), the association between clinical disease, SVs and virulence is much more complicated than originally thought. Recently our group showed that *G. parasuis* strain 174 (SV7) passaged in pigs can cause severe disease in pigs experimentally infected with the recovered strain [5]. Furthermore, we have already observed that clinical strains belonging to serovars 1, 4, 5, 13, 14 and 15, isolated from systemic sites of pigs with GD, were not capable of causing disease in controlled challenges using specific-pathogen-free piglets as an animal model. This reinforces that the prediction of virulence of a *G. parasuis* strain is complex and goes far beyond its capsular type.

*G. parasuis*, *Actinobacillus pleuropneumoniae* and *A. suis* are capable of acquiring iron from porcine transferrin for growth in low-iron environments such as the mucosal surface by expressing a transferrin (Tf) receptor that steals iron from porcine transferrin [8,9]. The strict host specificity of the Tf receptors limits the niche of the bacterial pathogens to a specific host that involved the coevolution of Tf and Tf receptors in a process that occurred over millions of years [10], suggesting that the surface Tf receptors likely arose in Gram-negative bacterial ancestors over 320 million years ago [11]. Thus, the receptors in the three porcine pathogens come from a common ancestor, which is why the overall diversity of the receptor proteins is distributed amongst the three species [9]. In addition to a common ancestry, the presence of efficient natural transformation systems in these species [12] provides ongoing opportunities for genetic exchange. As a consequence, vaccine antigens developed against *G. parasuis* will also be effective against *A. pleuropneumoniae* and *A. suis* since all three species are dependent upon the receptors for survival.

This iron-acquiring capability is mediated by two transferrin-binding proteins (Tbps). A surface lipoprotein named TbpB that is anchored to the outer membrane by fatty acyl chains on the N-terminal cysteine is able to extend from the surface with its long anchoring peptide region to specifically bind to the iron-loaded form of porcine transferrin (pTf) [13,14]. TbpB delivers the iron-loaded Tf to TbpA, an integral membrane protein, which then removes the iron molecule and transports it to the bacterial periplasm [15]. TbpB is crucial for *A. pleuropneumoniae* survival in pigs [8] and it is a protective antigen against *G. parasuis* [14,16,17] and *A. pleuropneumoniae* [18]. The sequence diversity of TbpB is distributed into three phylogenetic clusters with limited diversity in each cluster [9], suggesting that a maximum of three TbpBs would be required to induce an effective cross-protective response against all strains of the three species.

A non-binding-site-directed mutant of TbpB, named Y167A, when used as a vaccine antigen, protects pigs against lethal challenge using the Nagasaki (SV5) [14] and 174 (SV7) [17] strains, both of them belonging to cluster III of TbpB. These results demonstrate that TbpB^Y167A^ confers protection regardless of the capsular type of the *G. parasuis* strain, confirming that a TbpB-based vaccine has the ability to protect pigs against infection caused by all *G. parasuis* SVs and potentially against *A. pleuropneumonia* infection as well. The recent demonstration that TbpB^Y167A^ can confer protection and prevent colonization in an intranasal challenge model with *G. parasuis* [16] raises the question of whether it may be able to protect against natural infection by *A. pleuropneumoniae* if the neutralization of toxins is less important during the colonization phase.

Although the use of the TbpB protein as an antigen for a universal vaccine against *G. parasuis* has been consistently proven in experimental studies, only classical bacterin vaccines are commercially available to control GD. This type of vaccine does not provide broad heterologous protection [19,20] and, consequently, its effectiveness in complex production systems, such as nurseries housing animals from different origins, is very low. These poorly efficacious classical vaccines are predominantly used in Brazilian pig farming, which is likely why we were able to demonstrate the circulation of a large panel of well-defined *G. parasuis* serovars (1, 2, 4, 5, 12, 13, 14 and 15) in Brazilian pig farms including the circulation of nine new potential serovars not yet characterized [21]. To mitigate the development of GD caused by multiple *G. parasuis* serovars, the use of autogenous vaccines in combination with antibiotics throughout the nursery phase is a common practice in Brazilian farms.

In this study we demonstrate that *G. parasuis* SV7 is associated with clinical cases of GD in Brazil. Therefore, through a refined antigenic analysis (in vitro), we demonstrated that anti-TbpB IgG (cluster III) recognizes all TbpBs of all the analyzed strains including those belonging to cluster I. Finally, through an in vivo analysis, using a highly susceptible pig model, we demonstrated that the vaccine based on TbpB^Y167A^ is able to prevent GD caused by *G. parasuis* SV7 expressing cluster I TbpB. In contrast, a bacterin formulated with SV5 does not provide heterologous protection against SV7.

## 2. Materials and Methods

### 2.1. Glaesserella Parasuis Field Isolates

Twenty-seven isolates of *G. parasuis* were included in this study. The clinical samples were isolated by swabbing lungs, heart, meninges and joints of piglets aged 35 to 60 days presenting clinical signs of GD (high fever > 40.5 °C, coughing, abdominal breathing, swollen joints, lateral decubitus, trembling and paddling) from August 2016 to June 2018. Swabs were transported under refrigeration (4–8 °C) to the Laboratory of Advanced Microbiology and Immunology of the University of Passo Fundo, Brazil, and seeded on chocolate agar plates that were incubated on anaerobic jars at 37 °C for 24 to 36 h. Colonies with a morphology typical of *G. parasuis* were sampled and the bacterial identity and serovar classification was confirmed by multiplex PCR [22]. All confirmed isolates were cultured in PPLO broth supplemented with NAD (75 µg/mL) and glucose (2.5 mg/mL) to prepare the bacterial stocks that were stored in cryotubes at −80 °C. In addition, the Nagasaki (SV5) and 174 (SV7) reference strains were included in this study.

### 2.2. Virulence Prediction LS-PCR

Originally the 174 strain of *G. parasuis* SV7 was described as non-virulent [23]; however, we previously demonstrated that it can cause GD in colostrum-deprived piglets [17] as well as in conventional animals [16], thus we investigated whether the clinical strains identified as SV7 had virulence-associated trimeric autotransporters (vtaA) genes [24]. To do this, the genomic DNA (gDNA) was isolated from PPLO overnight cultures using the Wizard^®^ Genomic DNA Purification (Promega, Madison, WI, USA). The gDNA was quantified by nanophotometry (Implen, Munich, Germany), diluted to 20 ng/µL in ddH_2_O and stored at −80 °C until use. The PCR reaction using a leader-sequence-specific primer was carried out as previously described [24], and the PCR amplicons were analyzed by electrophoresis using 2% agarose gels.

### 2.3. TbpB Gene Amplification and Sequencing

The *tbpB* gene was PCR-amplified as previously reported [9] in a final volume of 20 µL, composed of 4 µL of 5X Phire Reaction Buffer, 0.4 mM of dNTPs, 400 nM of each primer (TbpB-F1 and TbpB-R), 0.4 µL of Phire Hot Start II DNA polymerase (ThermoFisher, Waltham, MA, USA) and 50 ng of gDNA. Cycling conditions were 98 °C for 30 s followed by 35 cycles of 98° C for 5 s, 55 °C for 5 s and 72 °C for 15 s, and a final extension at 72 °C for 1 min. The PCR products were analyzed by 1% agarose gel electrophoresis and then cleaned using Wizard SV Gel and PCR Clean-Up System (Promega, USA) following manufacturer’s instruction. The *tbpB* gene was sequenced by the Sanger method using previously reported primers [9].

### 2.4. Phylogenetic Analysis

The phylogenetic analysis was performed to evaluate the global diversity of the TbpBs amino acid sequences. Twenty-seven *tbpB* gene sequences from *G. parasuis* SV7, from our current study, and 66 previously published sequences [9] were concomitantly evaluated. The mature TbpB protein sequences were determined using SignalP 4.0 (http://www.cbs.dtu.dk/services/SignalP/, accessed on 24 November 2020) and the alignments generated using MAFFT v7 (https://mafft.Cbrc.jp/alignment/server/, accessed on 25 November 2020). The phylogenetic tree of maximum likelihood was build using PhyML43 and the substitution model WAG 44 with 100 bootstraps to evaluate the pattern of branching. The phylogenetic tree visualization and annotations were performed using FigTree.

### 2.5. TbpB Expression Analysis on Glaesserella Parasuis

The expression of the TbpB protein on the surface of the 27 clinical isolates of *G. parasuis* SV7 was evaluated in the presence of deferoxamine, which chelates iron molecules and thus induces the expression of TbpB. In brief, SV7 clinical isolates were cultivated in supplemented PPLO broth under constant agitation (250 rpm, Inova^®^ 40 NewBruswick, Brunswick, Germany) until reaching an optical density (OD) of 0.2 at 600 nm (Implen, Germany). Deferoxamine (80 µg/mL) was added to the PPLO broth and bacteria were further cultivated under the same conditions for 4 h. After that, bacteria were washed thrice with PBS and quantified by flow cytometry (BD FACSVerse, BD Biosciences^®^, San Jose, CA, USA), as previously described [25]. A total of 1 × 10^6^ bacteria were incubated with 200 µg of iron-loaded porcine transferrin (pTf) (First link Ltd., Birmingham, UK) labelled with fluorescein isothiocyanate (FITC, Sigma Aldrich, St. Louis, MO, USA) for 1 h at 37 °C. After this, bacteria were again washed thrice with PBS + 1% bovine serum albumin (BSA), suspended in 200 µL of PBS and then analyzed by flow cytometry. The relative expression of TbpB on the bacteria surface was estimated by calculating the percentage of bacteria gated in the P1 region with fluorescence (bacteria with associated pTf-FITC). The analysis was performed twice at two different times. Results are expressed as mean + standard deviation.

### 2.6. Capacity of Anti-TbpB^Y167A^ IgG in Recognizing Native TbpBs on Clinical Isolates of G. parasuis SV7

For this assay we used the clinical isolates of *G. parasuis* SV7 grown under iron-restriction conditions as described above. Then, 10 µL of complement-inactivated serum samples from pigs (*n* = 6) immunized with the TbpB^Y167A^ [17] were mixed with 1 × 10^7^ bacteria for 1 h at 37 °C. After that, bacteria were washed thrice with PBS and antibodies bound to native TbpBs were detected using goat anti-porcine IgG (H + L)-Phycoerythrin (PE) (Santa Cruz Biotech, Dallas, TX, USA) diluted 1:1000 in PBS + 1% BSA (1 h at 37 °C). The bacteria were subsequently washed thrice and suspended in 200 µL of PBS + 1% BSA and analyzed by flow cytometry. Serum from pigs immunized with a bacterin based on *G. parasuis* strain 174 (SV7) was used as a positive control. Serum from non-immunized SPF pigs of the same age was used as a negative control. Serum recognition was estimated by calculating the percentage of bacteria present in the P1 region with fluorescence (bacteria with associated anti-TbpB IgG). The analysis was performed twice at two different times. Results are expressed as mean + standard deviation.

### 2.7. In Vivo Evaluation of the Protective Capacity of the TbpB^Y167A^ (Cluster III) against G. Parasuis SV7 (TbpB Cluster I)

In order to assess whether pig anti-TbpB cluster III (mutant Y167A) IgGs were able to control *G. parasuis* SV7 (strain LM 360.18), which expresses a TbpB belonging to cluster I, we performed an immunization experiment and a controlled experimental challenge using pigs as an animal model. Twenty-four colostrum-deprived piglets (Agroceres PIC, Rio Claro, Brazil) were used in this experiment. The animals were produced by AFK Imunotech following the protocol already described by our group [17]. Twenty-one days after birth, piglets were weighed, ear tagged, divided into 4 homogeneous groups of 6 pigs and moved to a common infection room (without physical separation), equipped with plastic floor, temperature control (set at 22 °C), humidity control (80%), microbiological filter (GSI, Sunnyvale, CA, USA), and air recirculation (every 5 min). In addition, nasal and oropharyngeal swab samples were collected from all piglets and tested by PCR for *G. parasuis* [26], *A. pleuropneumoniae* [27], *Pasteurella multocida* [28], *Bordetella bronchiseptica* [29] and *Streptococcus suis* [30]. Serum samples were also tested for PCV2 [31]. All samples were negative for the presence of the described pathogens. On day 24, the piglets were immunized as illustrated in Figure 1, and 14 days after the second immunization (day 39) all pigs were anesthetized and intranasally challenged with 10^6^ *G. parasuis* strain LM 360.18 as described before [16].

### 2.8. Clinical Evaluation

Rectal temperatures and other clinical signs such as weakness, apathy, limping, sneeze, cough, dyspnea, lack of coordination, and/or loss of appetite were monitored once a day until the end of the study.

### 2.9. Vaccine Immunogenicity

To assess the antibody production during the immunization process, pig sera from all piglets were analyzed by Indirect ELISA. Three in-house ELISAs were used: (i) based on recombinant TbpB^Y167A^ protein; (ii) based on *G. parasuis* LM360.18 strain (SV7); and (iii) based on *G. parasuis* Nagasaki strain (SV5). Maxisorp plates (NUNC, Ocala, FL, USA) were used to prepare the ELISA with the purified TbpB^Y167A^ protein [17], which was used to analyze sera from Groups 1 (TbpB^Y167A^-based vaccine) and 4 (non-vaccinated). Polysorp plates (NUNC, USA) were used to prepare the ELISA with whole cells of *G. parasuis* [32], which was used to analyze samples from Groups 2 (bacterin based on SV7), 3 (bacterin based on SV5) and 4 (non-vaccinated). Briefly, 100 µL of serum samples serially diluted (from 1:100 to 1:102,400) in PBST (PBS + 0.05% Tween 20) containing 1% skim milk were added to the wells and incubated for 1 h at 37 °C. Then, wells were washed 3 times with PBST and 100 μL of goat anti-pig whole IgG peroxidase conjugated (Sigma-Aldrich, St. Louis, MO, USA) diluted 1:40,000 were added to the wells and incubated for one hour at 37 °C. The wells were washed again and the enzymatic reaction was developed with 3,3,5,5′-tetramethylbenzidine (Sigma-Adrich, USA) +0.06% H_2_O_2_ (Sigma-Adrich, USA). The plates were then incubated in the dark at 22 °C for 15 min and the reaction was stopped by adding 3 N HCl. Plates were read at 450 nm using a Synergy HI plate reader (Bio-Tek, Winooski, VT, USA). The results of the quantitative analysis were described as endpoint titers which are the reciprocal of the highest dilution that gave a positive OD reading (defined as at least two times greater that the OD values of the negative samples (pigs inoculated with PBS + Gel 02)). The IgG titers were expressed as Log10.

### 2.10. Ethical Statement

The experiment followed the guidelines of the Brazilian College of Animal Experimentation and was approved by the Committee for Ethical Use of Animals of the University of Passo Fundo (protocol no. 020/2020).

### 2.11. Statistical Analysis

Differences amongst groups were analyzed by two-way ANOVA followed by the Sidak post-test depending on the data normality assessed by the Kolmogorov–Smirnov and Levene tests. The comparative survival analysis illustrated was performed by Kaplan–Meier curve analysis. The results are reported as means ± SEM and *p*-values < 0.05 were considered to be significant.

## 3. Results and Discussion

### 3.1. Characterization and Geographical Distribution of SV7 Clinical Isolates

From August 2016 to June 2018, clinical isolates were obtained from pigs with presumptive GD and 27 were classified as SV7. The serovar identification was performed using a multiplex PCR as previously described [22]. As illustrated in Figure 2, the clinical isolates were from different farms located in six different Brazilian states: Minas Gerais (55%), Santa Catarina (22%), Paraná (12%), Rio Grande do Sul (4%), Goiás (4%) and Mato Grosso (3%) that comprise the most important swine-producing region (Central West and South regions) of Brazil.

Despite the original observations describing the 174 strain of *G. parasuis* SV7 as non-virulent [23], we demonstrated that it can cause GD in colostrum-deprived piglets [17] and in conventional animals [16]. Therefore, we investigated whether the clinical strains identified as SV7 had the leader sequence of the virulence-associated trimeric autotransporter (*vtaA*) genes found in disease-associated strains [24]. As illustrated in Figure 3B, the LS-PCR results indicated that the 27 clinical strains can be considered potentially virulent. In addition, we demonstrated that the 174 strain of *G. parasuis*, a reference strain for SV7, contains the same translocation domain that is used as a molecular marker of virulence. This result correlates with our clinical observations of piglets that were experimentally infected with the 174 strain of *G. parasuis* [16,17].

We recently demonstrated that the *G. parasuis* serovars 1, 2, 4, 5, 12, 13, 14 and 15 are well-distributed and associated with GD in Brazil [21]. Today this list has been expanded as we have demonstrated for the first time the presence of virulent strains of *G. parasuis* SV7 causing GD in Brazilian pig farms. The global circulation of virulent strains typed as SV7 has been demonstrated in different studies [33,34], which highlights the importance of this serovar as the causative agent of GD.

### 3.2. Phylogenetical Analysis of TbpB from Glaesserella Parasuis SV7

The *tbpB* gene from all clinical isolates of *G. parasuis* SV7 was sequenced and the complete, deduced amino acid sequence was aligned. All of the sequences were classified within cluster I and III, according to the strategy proposed for this protein [9], as illustrated in Figure 4.

Out of the 27 TbpBs from the SV7 clinical isolates, 24 (85.7%) were classified within cluster III and showed low amino acid variation as previously reported [9]. A similar study carried out with clinical strains of *G. parasuis* from Spain showed that approximately 45% of the strains belonged to cluster I (unpublished results) demonstrating that there may be regional differences in the distribution of TbpB variants. When comparing the sequences obtained in this study with other sequences already published [9], we observed that there are six phylogenetic subgroups within cluster III, with the amino acid variations located fundamentally in the loops found in the N lobe of the TbpB protein. We detected three strains expressing TbpB classified within cluster I (Figure 4), suggesting that these strains could potentially evade the antibody response induced by the TbpBY167A protein. To evaluate this hypothesis, we carried out two experiments: (i) first, we analyzed the antigenic characteristic of the IgG response induced with the TbpB^Y167A^ protein against the 27 strains of *G. parasuis* SV7; and (ii) second, we performed an in vivo experiment to assess whether the recognition predictions observed in the antigenicity analysis translated into clinical protection.

### 3.3. Antigenicity Analysis

*G. parasuis* and *A. pleuropneumoniae* use Tbps as the main system for iron uptake [8,35]. As expected, all clinical isolates grown in the presence of deferoxamine overexpressed TbpB on the membrane surface, which allowed us to perform the antigenicity analysis, as illustrated in Figure 5A. As illustrated in Figure 5B, pig anti-TbpB^Y167A^ IgG recognized the native TbpB expressed on clinical isolates regardless of their phylogenetic classification (I or III); although there are differences in the amino acid composition within the TbpBs from cluster I, II and III [9], there clearly are conserved epitopes that provide an opportunity for recognition by antisera from different TbpB variants. In practical terms, this in vitro analysis demonstrates that the TbpB^Y167A^-based vaccine is capable of inducing IgGs that recognize all the clinical strains of *G. parasuis* SV7 included in this study.

### 3.4. TbpB^Y167A^ Induces Protection against G. parasuis Strains That Express TbpB Cluster I

Although we demonstrated that the antibodies induced by the TbpB-based vaccine recognize all strains of *G. parasuis* SV7 (Figure 5), we wanted to verify whether the in vitro results translated into in vivo protection. To accomplish this objective, we performed an immunization experiment followed by a controlled challenge with a Brazilian clinical strain of *G. parasuis* SV7 (strain 360.18). In the design of this experiment, we included two other vaccines: i) a vaccine based on inactivated whole cells of *G. parasuis* SV7 strain LM 360.18 (positive control for protection against strain LM 360.18 infection—homologous protection); ii) a vaccine based on inactivated whole cells of *G. parasuis* SV5 strain Nagasaki (vaccine to assess heterologous protection between SV5 vs. SV7 capsular types).

As illustrated in Figure 6, all the vaccines induced the production of specific IgGs during the immunization protocol. An evident anamnestic response could be observed after the second immunization, which is a feature that was already observed in our previous studies [14,16,17,36], and related to the type of adjuvant used in the vaccine formulation.

In order to evaluate if the three experimental vaccines were capable of inducing a protective immune response against a lethal dose of the *G. parasuis* LM 360.18 strain, colostrum-deprived pigs were challenged with 1 × 10^6^ *G. parasuis* by the intranasal route, as previously described [16]. All the pigs immunized with the TbpB^Y167A^-based vaccine and those immunized with the vaccine based on inactivated whole cells of *G. parasuis* SV7 strain LM 360.18 survived for 14 days post challenge (experimental endpoint), achieving 100% protection, while all the pigs immunized with the vaccine based on inactivated whole cells of the *G. parasuis* SV5 Nagasaki strain or those inoculated with PBS-Gel 02 died or were euthanized during the post-challenge period, as illustrated in Figure 7. The rectal temperatures and the clinical symptom records are described in Appendix A.

Currently, the foundation for the prevention of GD is commercial vaccines that are based on chemically inactivated whole cells, and the protection induced by type of vaccine is thought to be directed against the extracellular capsular polysaccharide, which is very specific, thus little cross-protection is anticipated against strains expressing other capsular types [17]. This observation can be reinforced if we consider the theoretical basis of the main *G. parasuis* typing test that was used until 2015: the indirect hemagglutination, which uses specific antisera produced against the 15 reference serotypes of this agent [37]. Thus, although there are cross-reactivities between some serovars, such as between SV5 and SV12, among the others the reactions are practically non-existent.

Although it does not seem rational given the importance of GD, few studies on cross-protection between serovars have been carried out on pigs to date and the available results need to be interpreted with great caution because the animal model is not standardized (e.g., conventional pigs vs. specific-pathogen-free vs. colostrum-deprived pigs), and the virulence of the strains used in the challenges are also potentially different. The absence of heterologous protection between serovars 2 and 5 [19] and between serovar 12 and serovars 1, 4 and 5 [38] have already been demonstrated. In this study, we demonstrated that a vaccine based on *G. parasuis* SV5 does not induce cross-protection against serovar 7 (Figure 7), which reinforces the limited capacity of heterologous protection of classical vaccines based on inactivated whole *G. parasuis*.

In contrast, we demonstrated that the TbpB^Y167A^-based vaccine was fully protective against a strain expressing a TbpB from a different phylogenetic cluster, raising the question of whether this vaccine would be capable of providing protection from infection by any strain of *G. parasuis.* Since the TbpB variants are also distributed amongst strains of *A. pleuropneumoniae* and *A. suis* (Figure 4) [9], the prospect of a single vaccine capable of preventing infection from three porcine pathogens can be considered. However, it remains a challenge to provide experimental evidence that support a fully cross-protective response against all strains expressing different TbpB variants as it would be very costly using conventional approaches. Thus, attempts to use correlates of protection, as has been implemented in this study, are one approach, but it may be necessary to develop novel approaches such as generating barcoded strain libraries of variants to provide convincing experimental evidence from challenge experiments.

## 4. Conclusions

This study has provided convincing evidence that it is possible to induce a cross-protective response with the Y167A mutant TbpB against a strain expressing a TbpB from a different phylogenetic cluster, raising the prospect of a single vaccine antigen providing complete protection from three important pathogens that express variants from all three phylogenetic clusters.

## Figures and Tables

**Figure 1 pathogens-11-00766-f001:**
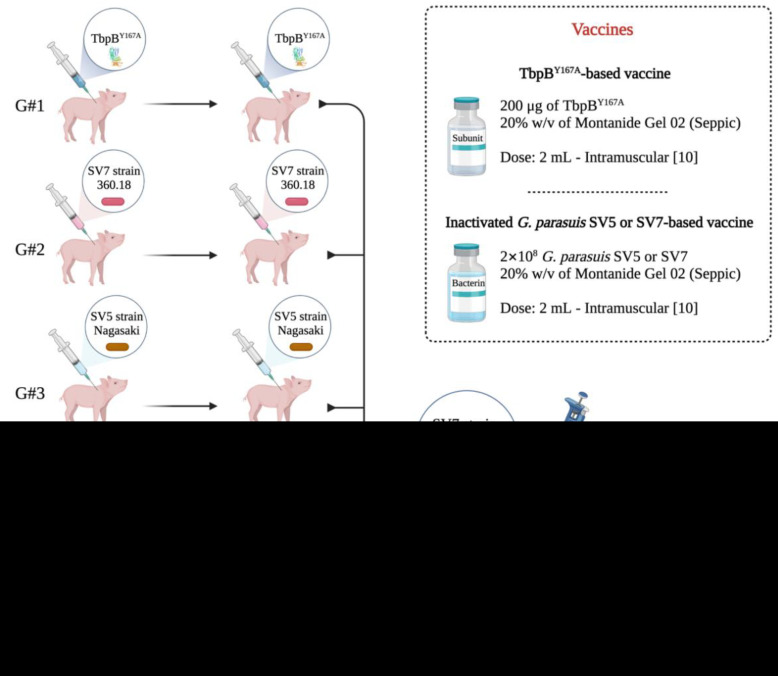
Experimental design of immunization and experimental infection with *Glaesserella parasuis* SV7 (*n* = 6 piglets per group). Group 1 (**G#1**): immunized twice with the TbpB^Y167A^-based vaccine. Group 2 (**G#2**): immunized twice with the inactivated *G. parasuis* SV7 (strain LM360.18)-based vaccine. Group 3 (**G#3**): immunized twice with the *G. parasuis* SV5 (strain Nagasaki)-based vaccine. Group 4 (**G#4**): inoculated twice with PBS + 20% (*w*/*v*) Montanide Gel 02 adjuvant. Fourteen days after re-vaccination, all pigs were challenged by the intranasal route with 10^6^ *G. parasuis* SV7 (strain LM 360.18), and clinically monitored once a day until the end of the experiment. The description of the vaccine composition is illustrated in the figure. This figure was created with BioRender.com (accessed on 17 May 2022).

**Figure 2 pathogens-11-00766-f002:**
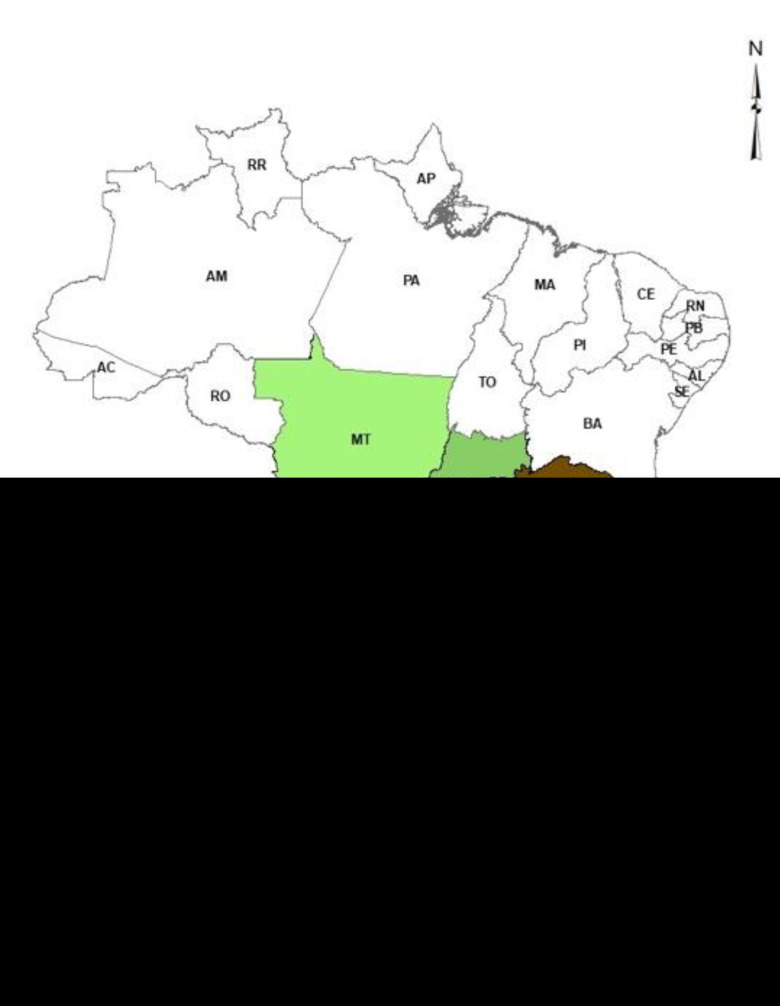
Geographic distribution of *Glaesserella parasuis* serovars 7 in Brazil.

**Figure 3 pathogens-11-00766-f003:**
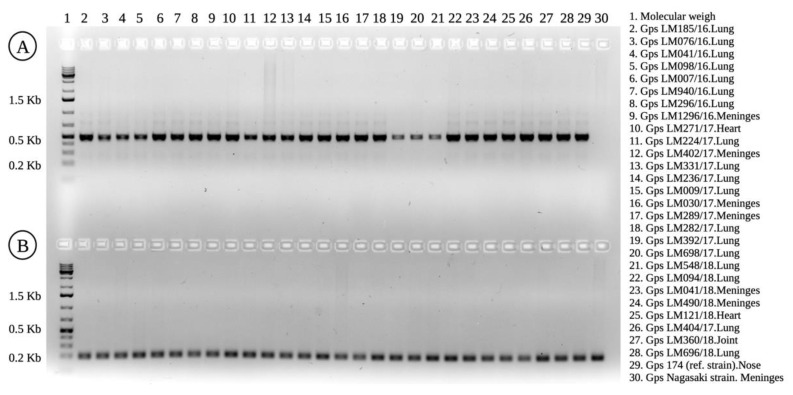
Molecular analysis of *Glaesserella parasuis* in relation to capsular type and virulence prediction. (**A**) Uniplex PCR using FunQ Forward and Reverse primers, which amplified serovar 7 (product of 490 bp) (lanes 2–29). The 174 strain (reference for SV7) was used as a positive control (lane 29) and Nagasaki strain (SV5) as a negative control (lane 30). (**B**) PCR results based on the leader sequence of the *vtaA* genes of *G. parasuis*. A PCR product of 200 bp was obtained when the primers AV1-F and V1-R were used.

**Figure 4 pathogens-11-00766-f004:**
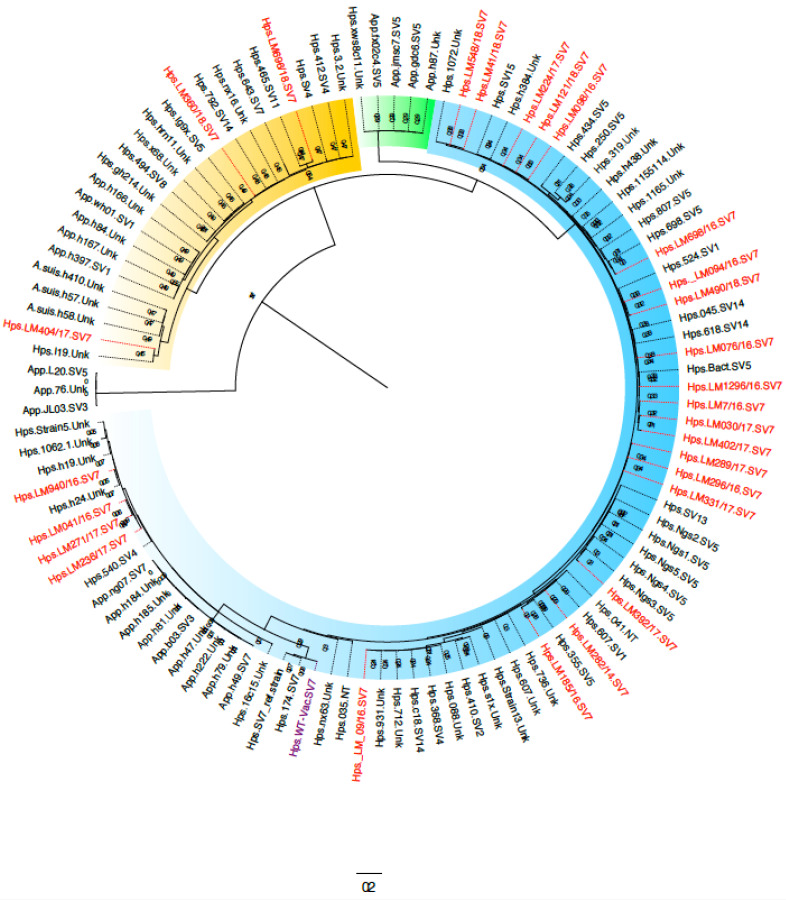
Sequence diversity of TbpBs from porcine pathogens. Maximum likelihood tree demonstrating the overall diversity of TbpBs from *G. parasuis, A. pleuropneumoniae* and *A. suis*. Leaf labels identify the strains from which TbpB sequences were obtained and indicate their species and serovar, if known (NT = Nontypeable, Unk = Unknown). The new sequences described in this study are labelled in red. The sequences are clustered into three main groups (Group 1 = yellow background, Group 2 = green background, Group 3 = blue background) with high confidence. The branch support values are displayed.

**Figure 5 pathogens-11-00766-f005:**
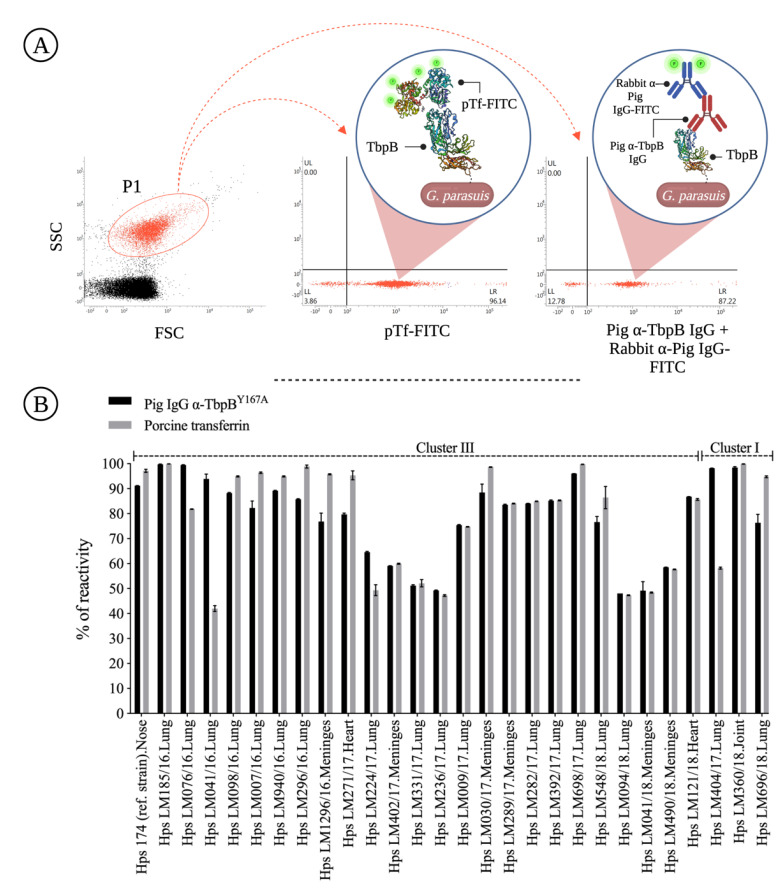
Porcine anti-TbpB^Y167A^ IgG antigenicity analysis against clinical strains of *Glaesserella parasuis* SV7. (**A**) Cytometric strategy to analyze the expression of TbpB, as well as to assess the ability of porcine anti-TbpB^Y167A^ IgGs to recognize native TbpB. (**B**) Results of the antigenicity experiment, where porcine anti-TbpB^Y167A^ IgGs recognize native TbpB belonging to clusters III and I expressed on the surface of 27 clinical strains of *G. parasuis* SV7. The percentage (%) of reactivity refers to the % of *G. parasuis* gated in the P1 region (**A**) with pTf and/or antibodies associated on its surface.

**Figure 6 pathogens-11-00766-f006:**
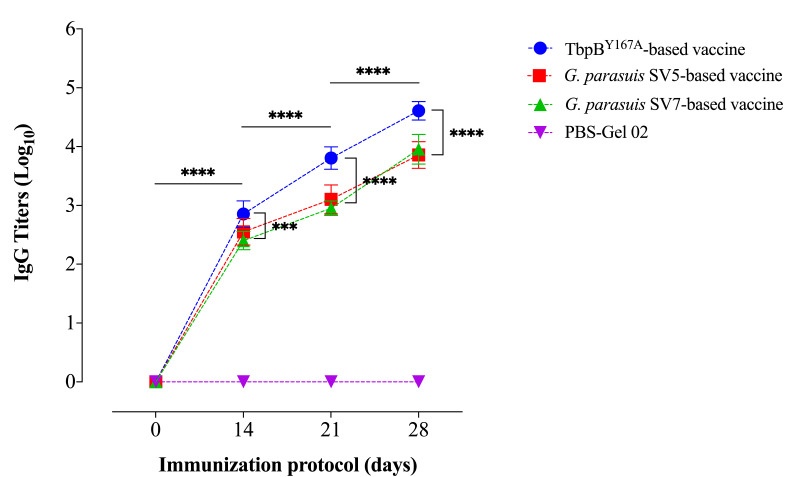
Serum IgG response against TbpB^Y167A^, whole *G. parasuis* serovar 5 and 7 in piglets immunized during the nursery phase (as depicted on Figure 1). The IgG titer (mean ± SD) is expressed as log_10_. Multiple comparisons (two-way ANOVA) were performed by first comparing the evolution of IgG titers at different times of the immunization protocol within each experimental group, and subsequently, comparing the IgG titers observed at different times between the different experimental groups. The serum of the control animals was evaluated in the three different ELISAs, and the results were negative for all. Statistical differences are indicated in the figure by asterisks (*** *p* < 0.001, **** *p* < 0.0001).

**Figure 7 pathogens-11-00766-f007:**
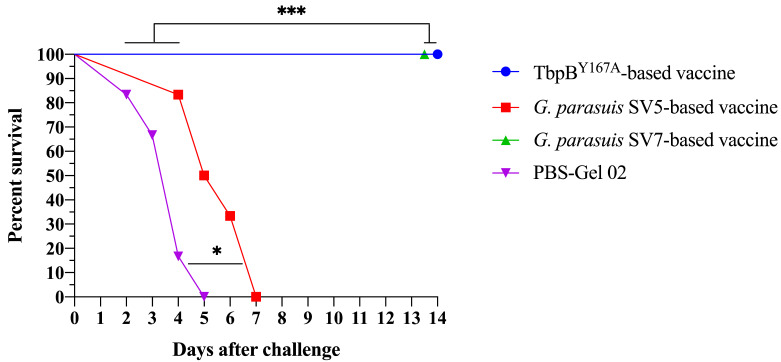
Survival rates of pigs challenged with *Glaesserella parasuis* SV7 LM 360.18 strain. Groups of pigs were inoculated with PBS-Gel 02 (control) or vaccinated with TbpB^Y167A^-based vaccine, *G. parasuis* SV5-based vaccine, or *G. parasuis* SV7-based vaccine. The pigs were challenged by intranasal inoculation with 1 × 10^7^ *G. parasuis* LM 360.18 strain and were monitored for clinical signs and symptoms throughout the 14 days of the experiment. Statistical differences are indicated in the figure by asterisks (* *p* < 0.05 and *** *p* < 0.001).

## Data Availability

Not applicable.

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
