# Peer review of "TbpBY167A-Based Vaccine Can Protect Pigs against Glässer’s Disease Triggered by Glaesserella parasuis SV7 Expressing TbpB Cluster I"

_pathogens, 2022, doi:10.3390/pathogens11070766_

Round 1
Reviewer 1 Report
Major comments:
This manuscript by Simone Ramos Prigol et al. provides some interesting information about TBPY167A-based vaccine, especially the protective role of this vaccine in G. parasuis SV7 infections. However, I have some concern in the current manuscript, as follows:
1, TBPY167A-based vaccine has been used all over the manuscript, but the information of TBPY167A-based vaccine is missed.
2, The different clusters of TBP by phylogenetic analysis is the first time to be determined or has been reported? If it is the first time to be determined by this manuscript, there were a lot of things going on about this method, which included: 1) Phylogenetic analysis could show the similarity of TBP from different strains, but it is not accurate as the homology analysis. Furthermore, TBP as an protective antigen, its extracellular domain is the key for induction of immune response. So it is very hard to tell about similarity of TBP subfragment, like extracellular domain. Anyay, for me only a phylogenetic analysis is not enough to conclude similarity of a protein or its protective domain among different strains; 2) Line 69-72, the cluster III of TBP had been used intro without ref; 3) According to figure 4, some SV7 stains were branched into cluster I, some in the other cluster, i.e cluster is not specific to any serotype of G. parasuis strain. But, how to understand the meaning of the clusters, like link to protection among different starins, even protection among different bacteria (G. parasuis vs APP?)? It must be addressed clearly; 4) Line 299-302, regarding missing information of TBPY167A-based vaccine and expression of cluster I TBP, how to come up with this suggestion, please give more information.
3, Line 68-79, for me it is very hard to understand those information? I guess, TBPY167A-based vaccine has the pan-protect role in not only G. parasuis, but also APP. If so, which kind of TBPY167A-based vaccine has this role?
4, Line 178-183, it is very hard to understand. The subtitle about protective role of TBPY167A (cluster III) in G. parasuis SV7 strain with cluster I TBP, somehow, cluster and serotype complicated the understanding, at leas for me. But the real point is that which kind of cluster TBP was expressed by G. parasuis SV7 strain?
5, In conclusion, Line 405-409, which kinds of data can support this conclusion, please give the explanation.
Minor comments:
1, Ref 7, are you sure that pigs had been challenged with 174(SV7)?
2, Figure 3, it is understandable about vtaA gene, but how to understand FunQ gene.
3, Figure 5A, I don’t like one gate for two different analyses; Figure 5B, how to understand the reactivity, like more than 30%, is something missed?
Author Response
Thank you for your review and the concise summary of the merits of the manuscript. We have answered each of your points below.- TBPY167A-based vaccine has been used all over the manuscript, but the information of TBPY167A-based vaccine is missed.
- R: Apology for the confusion. We have made substantive changes to the Introduction section so that a greater appreciation of the vaccine potential of the TbpBY167A-based vaccine should readily be understood.
- The different clusters of TBP by phylogenetic analysis is the first time to be determined or has been reported? If it is the first time to be determined by this manuscript, there were a lot of things going on about this method, which included: 1) Phylogenetic analysis could show the similarity of TBP from different strains, but it is not accurate as the homology analysis. Furthermore, TBP as an protective antigen, its extracellular domain is the key for induction of immune response. So it is very hard to tell about similarity of TBP subfragment, like extracellular domain. Anyay, for me only a phylogenetic analysis is not enough to conclude similarity of a protein or its protective domain among different strains; 2) Line 69-72, the cluster III of TBP had been used intro without ref; 3) According to figure 4, some SV7 stains were branched into cluster I, some in the other cluster,ecluster is not specific to any serotype of G. parasuis strain. But, how to understand the meaning of the clusters, like link to protection among different starins, even protection among different bacteria (G. parasuis vs APP?)? It must be addressed clearly; 4) Line 299-302, regarding missing information of TBPY167A-based vaccine and expression of cluster I TBP, how to come up with this suggestion, please give more information.
- R: The substantive changes to the introduction should address many of your expressed concerns about the protective capabilities of the receptor proteins, the phylogenetic clusters and their independence of capsular type or species. The description also clarifies that the TbpB protein is almost entirely extracellular as it is only anchored in the outer membrane by its fatty acylated N-terminal cysteine.
- Line 68-79, for me it is very hard to understand those information? I guess, TBPY167A-based vaccine has the pan-protect role in not only G. parasuis, but also APP. If so, which kind of TBPY167A-based vaccine has this role?
- R: This section merely states that the Y167A TbpB can induce a protective response against strains that express a TbpB of the same phylogenetic cluster regardless of the capsular type, and that this protective response against the TbpB would also apply to A. pleuropneumoniae.
- Line 178-183, it is very hard to understand. The subtitle about protective role of TBPY167A (cluster III) in G. parasuis SV7 strain with cluster I TBP, somehow, cluster and serotype complicated the understanding, at leas for me. But the real point is that which kind of cluster TBP was expressed by G. parasuis SV7 strain?
- R: I hope that the revised introduction will provide a better understanding of the sequence diversity so that this now makes sense to you. Basically the TbpBs within a phylogenetic cluster are quite similar in sequence and anticipated to induce a cross-protective response to strains expressing a TbpB within that cluster. The significance of this section is the demonstration of protection against strains expressing a TbpB from a different cluster.
- In conclusion, Line 405-409, which kinds of data can support this conclusion, please give the explanation.
- R: Hopefully this will now be obvious due to the changes made in the revised manuscript.
- Ref 7, are you sure that pigs had been challenged with 174(SV7)?
- R: Yes.
-
Figure 3, it is understandable about vtaA gene, but how to understand FunQ gene
- R: The PCR primers (FunQ gene) are unique to the serovar 7 capsular locus.
- Figure 5A, I don’t like one gate for two different analyses; Figure 5B, how to understand the reactivity, like more than 30%, is something missed?
- R: I respectively disagree. This illustrates that the first gate is used for two separate analyses (experiments) with illustrations of what is being used to label the cells.
Reviewer 2 Report
Dear Editor,
Please find my comments for “Ramos Prigol et al.” original article. Overall, the subject is interesting and the article well-presented. I have only moderate remarks. Have an excellent day.
All the best
François Meurens
General comments:
In this paper, Ramos Prigol et al. analyzed the genetic and immunological properties of the TbpB protein, a surface lipoprotein binding specifically to the iron-loaded form of porcine transferrin (pTf). This protein is naturally expressed in at least 27 different clinical isolates of Glaesserella parasuis, the etiological agent of Glässer’s disease (GD), as serovar 7 and isolated from systemic lesions from pigs having GD. Authors performed phylogentic analyses to compare the strains. Then, they analyzed cross-reactivity and cross-protection (homologous challenge compared to heterologous) using specific pathogen free pigs. Overall, the article is interesting and quite well-written. Please find below a detailed list of my moderate remarks/suggestions.
Strengths: Interesting and useful subject, original, and quite well-written even if English can be improved (several small mistakes and typos).
Weakness: More details about TbpB are needed in the abstract and the introduction. There is sometimes confusion between G. parasuis, A. pleuropneumoniae and A. suis. Authors could improve the article for that specific point.
Major
/
Moderate
-More context about TbpB is needed in the abstract too (as it is the case L65).
-L25: Authors should explain already here why it is protective. Where this information is coming from.
-L31: The sentence about the surviving pigs is quite confusing when we know that the pigs from G3 in Fig7 are not surviving. Please modify to remove all the possible confusion.
-L42: Reference 1, more general and older references about GD are also available. Please add some.
-L44-45: If the sow has been infected or vaccinated only… Please add the following reference about maternal immunity transfer (doi: 10.1016/j.dci.2008.07.007).
-L50-51: Please add the following references about co-infections and their molecular consequences doi: 10.1186/s13567-020-00807-8
-L65-79: The link between the two bacteria is not always clear, authors moving from one bacterium to the other. This is a general remark for the article in fact. The reader is not always which bacterium the authors are talking about.
-L81: What is a classical vaccine for the authors? Please clarify.
-L94: In which sense these pigs are highly susceptible, please clarify.
-Please described the SPF pigs. For which pathogens are they tested? And how?
-L102: Please mention these “typical” signs.
-L187: Why without physical separation? It can be an issue just after the inoculation of the bacteria during the challenge, sneeze… What is the rational?
-In Fig. 1, vaccination could be replaced by immunization.
-Please describe with more details the intranasal administration (volume…), L200.
-Fig4 is too small, it is really difficult to assess it.
-Around L372: Where is the discussion starting? There is no “discussion” title? That section is needed.
-L383-385: This is very frequently the case, in fact for most of the pathogens, see PRRSV studies or influenza virus studies.
-Figure 5: How many repetitions for each strains? No statistical comparisons?
-There is no real mock control group, non-immunized and unchallenged pigs? Please justify
Author Response
Dear prof. Meurens
Thank you for your review and for the concise summary of the merits of the manuscript. We have answered each of your points below.
Point 1: More details about TbpB are needed in the abstract and the introduction. There is sometimes confusion between G. parasuis, A. pleuropneumoniae and A. suis. Authors could improve the article for that specific point.
Response 1: Thanks. We have made substantial revisions particularly to the introduction section that will hopefully clarify the confusion. The abstract is already over the allowed limit so we could only add details on TbpB there if the journal allows it.
Point 2: L31: The sentence about the surviving pigs is quite confusing when we know that the pigs from G3 in Fig7 are not surviving. Please modify to remove all the possible confusion.
Response 2: Thanks. We have made substantial revisions particularly to the introduction section that will hopefully clarify the confusion.
Point 3: L42: Reference 1, more general and older references about GD are also available. Please add some.
Response 3: Thanks. The new reference was included.
Point 4: L44-45: If the sow has been infected or vaccinated only… Please add the following reference about maternal immunity transfer (doi: 10.1016/j.dci.2008.07.007).
Response 4: Thanks. The new reference was included.
Point 5: L50-51: Please add the following references about co-infections and their molecular consequences doi: 10.1186/s13567-020-00807-8
Response 5: Thanks. The new reference was included.
Point 6: L65-79: The link between the two bacteria is not always clear, authors moving from one bacterium to the other. This is a general remark for the article in fact. The reader is not always which bacterium the authors are talking about.
Response 6: Thanks. We have made substantial revisions that will hopefully clarify the confusion.
Point 7: L81: What is a classical vaccine for the authors? Please clarify.
Response 7: Classical bacterin vaccine.
Point 8: L94: In which sense these pigs are highly susceptible, please clarify.
Response 8: They are pigs free from maternal immunity (deprived of porcine colostrum) and free from respiratory pathogens that normally colonize the mucous membranes of pigs, such as Glaesserella parasuis, Pasteurella multocida, Actinobacillus suis, Bordetella bronchiseptica and Streptococcus suis.
Point 9: Please described the SPF pigs. For which pathogens are they tested? And how?
Response 9: Thanks. We have included this information in M&M section.
Point 10: L102: Please mention these “typical” signs.
Response 10: Thanks. We have included this information in M&M section.
Point 11: L187: Why without physical separation? It can be an issue just after the inoculation of the bacteria during the challenge, sneeze… What is the rational?
Response 11: The infection room of our bio-controlled experimental unit does not have separate pens. It is a single room of 60m2. In this case, all animals share the same environment, including coming into contact with animals that develop clinical disease. This aspect is interesting because it mimics what actually happens in a nursery unit. Although we cannot measure the burden of naturally transmitted G. parasuis between animals, it is clear (Fig. 7) that if the vaccine induces clinical protection piglets do not get sick even in contact with other sick animals.
Point 12: Please describe with more details the intranasal administration (volume…), L200.
Response: Thanks. It is described in the reference #16.
Point 13: Around L372: Where is the discussion starting? There is no “discussion” title? That section is needed.
Response: Line 265 - Results and Discussion.
Point 14: Figure 5: How many repetitions for each strains? No statistical comparisons?
Response: Each strain was analyzed twice at two different times. Results are expressed as mean + standard deviation. This information was included in the M&M section.
Point 15: There is no real mock control group, non-immunized and unchallenged pigs? Please justify
Response 15: Thanks for this observation. In this study, we had a limitation in the number of animals, and therefore, the inclusion of this physiological group was not possible. However, we fully believe that the absence of this group did not compromise the value of the findings obtained in this study; since it was possible to reproduce the disease in unvaccinated animals and to obtain protection in some vaccinated groups.
Round 2
Reviewer 1 Report
I don't have further comments to the current manuscript.
Reviewer 2 Report
The article has been well-improved and can, in my opinion, be accepted.